# Peroxisome Dysfunction and Steatotic Liver Disease

**DOI:** 10.3390/ijms26178303

**Published:** 2025-08-27

**Authors:** Navia Vinoy, Gary Huang, Daniel F. Wallace, V. Nathan Subramaniam

**Affiliations:** 1Hepatogenomics Research Group, Queensland University of Technology (QUT), Brisbane, QLD 4059, Australia; navia.vinoy@hdr.qut.edu.au (N.V.); gary.huang@connect.qut.edu.au (G.H.); 2Centre for Genomics and Personalised Health, School of Biomedical Sciences, Queensland University of Technology (QUT), Brisbane, QLD 4059, Australia; d5.wallace@qut.edu.au; 3Metallogenomics Laboratory, Queensland University of Technology (QUT), Brisbane, QLD 4059, Australia

**Keywords:** metabolic dysfunction-associated steatotic liver disease, liver disease, peroxisomes, beta-oxidation, fatty acid synthesis, fatty acid oxidation

## Abstract

Peroxisomes are cellular organelles involved in multiple metabolic processes, including lipid oxidation, lipid synthesis, and the metabolism of reactive oxygen species. Peroxisomal disorders arise from defects in peroxisomal biogenesis or peroxisomal enzymes. Patients with severe peroxisomal disorders often present with a range of distinctive physical features and congenital malformations, such as neuronal migration defects, renal cysts, and bony stippling in the patellae and long bones. Liver disease has also been reported in some patients with peroxisomal biogenesis disorders, although the exact molecular mechanisms underlying its development remain unclear. Metabolic dysfunction-associated steatotic liver disease (MASLD) is now recognised as one of the most prevalent causes of chronic liver disease globally, due to its widespread incidence and potential for serious complications. This review aims to highlight the possible involvement of peroxisomal defects in the pathogenesis of MASLD.

## 1. Introduction

Peroxisomes are microbodies, organelles that are bound by a single membrane [1]. They contain at least 50 diverse enzymes that are essential for various metabolic pathways [2]. Peroxisomes play a significant role in metabolic pathways, contributing to reactive oxygen species (ROS) detoxification, ether-phospholipid biosynthesis, various aspects of lipid metabolism, particularly fatty acid oxidation process, and bile acid biosynthesis in the liver [3]. Hepatocytes are among the most peroxisome-rich cells in the body, containing between 300 and 600 peroxisomes [4]. Peroxisomal dysfunction has been linked with a range of liver diseases, from steatosis and cholestasis to progressive liver failure [5,6]. Moreover, inherited peroxisomal abnormalities, such as Zellweger spectrum disorders, often present with significant liver dysfunction, underpinning the close association between peroxisomes and liver health [7]. Understanding the association between peroxisomal function and liver health is essential for unravelling the pathophysiology of these conditions.

Peroxisomal biogenesis is the process by which peroxisomes are created and maintained as described in Figure 1. It is a process driven by peroxins, proteins that are encoded by *PEX* genes. At least 18 distinct peroxins have been identified in humans, each playing crucial roles in peroxisomal membrane assembly, matrix protein import, division and proliferation, or quality control. As such, any genetic defects could affect peroxisome structure and function. The three key steps of peroxisomal biogenesis are: (a) peroxisomal membrane assembly, (b) peroxisomal matrix protein import, and (c) growth and division of the organelles [8].

The initial stage of de novo peroxisomal biogenesis involves the import of peroxisomal membrane proteins (PMPs) by PEX3, PEX16 and PEX19 [9], and matrix proteins by PEX5, PEX7, and the PEX13-PEX14 docking complex, into pre-peroxisomal structures [10]. These pre-peroxisomal structures originate from either the endoplasmic reticulum (ER) or the mitochondria, and are formed with the assistance of PEX3, PEX16 and PEX19—resulting in ER-derived pre-peroxisomal vesicles (ERDppVs) or mitochondrially derived pre-peroxisomal vesicles (MDppVs), respectively [11]. The vesicles then fuse PEX3-enriched sites to form early peroxisomes, which mature with further PMP import [12]. The PEX11 family of proteins (PEX11α, PEX11β, and PEX11γ) regulates peroxisomal abundance and size through elongation and fission [13], While PEX2, PEX10, and PEX12 initiate the monoubiquitylation of PEX5, essential for its recycling, PEX1 and PEX6 anchored to the peroxisomal membrane by PEX26 facilitate the adenosine triphosphate (ATP)-driven extraction of PEX5 from the membrane, allowing the recycling of import receptors back into the cytosol for reuse [14,15].

## 2. Role of Peroxisomes in Metabolic Pathways in the Liver

The liver is the organ that has received the most attention in research regarding the functionality of peroxisomes [16]. In the liver, peroxisomes are involved in fatty acid oxidation, glyoxylate detoxification, synthesis of bile acids, ether lipids, and cholesterol [17]. They are named “peroxisomes” due to their involvement in the production and breakdown of hydrogen peroxide (H_2_O_2_) [18]. Additionally, peroxisomes are crucial for the oxidation process of other organic substances including amino acids, fatty acids, and uric acid [17]. The peroxisomal enzyme catalase, plays a vital role in detoxifying H_2_O_2_, by breaking it down into water and oxygen [19]. This suggests that peroxisomal depletion might affect metabolism and affect liver function.

### 2.1. Fatty Acid Oxidation

In peroxisomes, most fatty acids are metabolised by β-oxidation [20]. Peroxisomal β-oxidation is responsible for breaking down very long-chain fatty acid (VLCFAs), di- and trihydroxycholestanoic acid (DHCA and THCA), and certain polyunsaturated fatty acids with the help of the rate-limiting enzyme acyl-coenzyme A oxidase (ACOX1) [21]. Peroxisomes also employ α-oxidation to metabolise branched-chain fatty acids that are unable to directly undergo β-oxidation, such as phytanic acid into pristanic acid due to its β-methyl branch. As such, defects in the α-oxidation of fatty acids can lead to the accumulation of phytanic acid, resulting in Refsum disease [22].

### 2.2. Ether Lipid Synthesis

Ether lipid synthesis is initiated in the peroxisome and completed in the ER [23]. Ether lipids contribute to forming structural components of cell membranes, and help regulate cellular signalling [24], cell differentiation [25] and membrane trafficking [26]. Ether lipids are also involved in the antioxidant defence mechanism [27]. Defects in peroxisomes lead to a deficiency in ether lipids, which is linked to the peroxisomal disorders Rhizomelic Chondrodysplasia Punctata (RCDP) and Zellweger syndrome (ZSD) [28].

### 2.3. ROS Regulation

Peroxisomes play a significant role in detoxifying ROS within cells [29]. H_2_O_2_ and superoxide (O_2_^−^) are ROS generated as byproducts from FAD-dependent oxidoreductases during aerobic metabolic processes such as bile acid synthesis or β-oxidation, due to the partial reduction of oxygen [30]. Catalase is a peroxisomal enzyme that protects cells from damage by converting H_2_O_2_ into oxygen and water [31]. Given its involvement in H_2_O_2_ homeostasis, impaired peroxisomal function could result in dysregulated oxidative stress and liver damage. A disruption in H_2_O_2_ homeostasis has been shown to contribute to several diseases such as cancer, diabetes, obesity, and renal injury [32].

### 2.4. Glyoxylate Detoxification

Glyoxylate is another toxic metabolic that is detoxified by peroxisomes [33]. Glycolate oxidase (GO) is a liver-specific peroxisomal enzyme encoded by the Hydroxyacid Oxidase 1 gene (*HAO1*) that is responsible for oxidising glycolate, sourced from the diet or endogenously through amino acid metabolism, into glyoxylate [34]. This is a process that produces H_2_O_2_. Peroxisomal defects could lead to glyoxylate accumulation, which is associated with primary hyperoxaluria type 1 [35].

### 2.5. Cholesterol and Bile Acid Synthesis

Peroxisomes are involved in the biosynthesis of cholesterol [36]. Numerous investigations attribute the initial pre-squalene stages, as well as subsequent stages of cholesterol biosynthesis to peroxisomes [37]. Apart from 3-hydroxy-3-methylglutaryl-coenzyme A reductase (HMGCR), the enzymes within the pre-squalene segment possess functional peroxisomal targeting signals (PTS) responsible for facilitating their translocation into the peroxisomal matrix [38]. It has been demonstrated that acetyl-CoA produced from peroxisomal β-oxidation of VLCFAs (e.g., 1,2,3,4-^13^C_4_] docosanoate) and dicarboxylic acids (e.g., [U-^13^C_12_] dodecanedioate) is directed towards cholesterol biosynthesis within peroxisomes [39]. Peroxisomal dysfunction could impair cholesterol transport leading to its accumulation in lysosomes and eventual cell death. This can trigger several diseases such as Gaucher’s disease, Niemann-Pick disease, and other lysosomal storage disorders [40].

Peroxisomes are also involved in the biosynthesis of bile acids as the peroxisomal β-oxidation process is required to produce fully developed C24-bile acids from C27-bile acid intermediates [41]. This process is facilitated by the peroxisomal enzyme bile acid-CoA:amino acid N-acyltransferase (BAAT), which conjugates bile acid to glycine or taurine [42,43]. Peroxisomal disorders such as ZSDs are associated with bile acid abnormalities, which can lead to growth impairment, liver disease, and other complications [44,45].

## 3. Crosstalk Between Peroxisomes and Other Organelles

Interaction with lysosomes, lipid droplets, ER, and mitochondria is necessary for peroxisomes to carry out their role in cellular metabolism [46]. These inter-organelle communications enable diverse metabolic pathways such as lipid biosynthesis, fatty acid oxidation, and redox balance [47]. Disruption in these inter-organelle communications can lead to compromised metabolic regulation and play a contributory role in the pathogenesis of several metabolic disorders [48].

### 3.1. Peroxisome-Lysosome Interaction

Dynamic membrane interactions between peroxisomes and lysosomes are mediated by the binding of lysosomal synaptotagmin VII (SYT7) with phosphatidylinositol 4,5-bisphosphate (PI(4,5)P2) on the peroxisomal membrane [49]. These interactions are enhanced in the presence of low-density lipoprotein (LDL)-cholesterol, to facilitate cholesterol transfer from lysosomes to peroxisomes [49]. Proper membrane interactions are crucial for lipid homeostasis, and disruption of critical peroxisome genes can result in lysosomal cholesterol accumulation.

### 3.2. Peroxisome-Lipid Droplet Interaction

Lipid droplets (LDs) are responsible for the storage of neutral lipids and facilitating lipid exchange with other organelles, including peroxisomes [50], ER [51], and mitochondria [52]. LD-peroxisome interactions, or lipid trafficking from LD to peroxisomes, are mediated by M1 spastin, localised to the LD surface, tethering with the peroxisomal transporter, ATP-binding cassette subfamily D member 1 (ABCD1) [53]. Disruption of M1 spastin or ABCD1 can impair lipid transfer, compromising the function of LDs and peroxisomes [54].

### 3.3. Peroxisome-ER Interaction

Peroxisome-ER interaction facilitates the transfer and metabolism of VLCFAs in peroxisomes [55]. VLCFAs are synthesised in the ER through the elongation of shorter-chain FAs [56]. The transfer of ER elongated acyl-CoAs to the peroxisome, is mediated by the tethering of ER-resident protein vesicle-associated membrane protein-associated protein B (VAPB) with peroxisomal acyl-CoA binding domain-containing protein 5 (ACBD5) [57,58]. ACBD5 deficiency leads to reduced ER-peroxisome contact, accumulation of VLCFA-containing phospholipids, increased peroxisomal mobility, and impaired peroxisomal membrane growth [59].

### 3.4. Peroxisome-Mitochondrion Interaction

Mitochondria and peroxisomes collaborate in lipid metabolism, particularly in β-oxidation of VLCFAs [60]. VLCFAs are initially chain-shortened in peroxisomes, and the resulting acyl-CoA are shuttled to the mitochondria for complete oxidation [60]. Defects in peroxisomal function can result in VLCFA accumulation and consequently impair mitochondrial integrity and function [61]. Mitochondria can control peroxisome biogenesis through various mechanisms, such as the sharing of fission machinery [62], vesicle contribution for peroxisomal precursor formation [11], and metabolic coordination [63]. This may even result in the mislocalization of peroxisomal proteins to the mitochondria, further disrupting cellular processes [64].

## 4. Peroxisomal Disorders

Peroxisomal disorders are rare hereditary conditions caused by defects in peroxisomal assembly, biogenesis, and function [65].X-linked adrenoleukodystrophy (X-ALD), is the most prevailing peroxisomal disorder with an estimated incidence ranging from 1 in 14,000 to 1 in 17,000 male births [66]. X-ALD mainly affects males with the full neurological phenotype due to its X-linked inheritance, while females can have milder symptoms [67]. In contrast, ZSD is the most frequent peroxisomal disorder diagnosed in early infancy with an estimated incidence of approximately 1 in 50,000 to 100,000 live births in the US [68]. However, incidence rates differ by region—increasing to about 1 in 12,000 in parts of Quebec and falling to around 1 in 500,000 in Japan [69]. The overall incidence of peroxisomal disorders is approximated at 1 in 50,000 to 100,000 live births [69]. They can be classified into two categories: single enzyme deficiencies or PBDs [70]. Single enzyme deficiencies are caused by mutations in genes that encode specific peroxisomal enzymes [71]. Single enzyme deficiency-associated disorders include: (1) fatty acid oxidation-associated disorders; (2) ether lipid synthesis disorder; and (3) glyoxylate peroxisomal disorder [72].

PBDs, on the other hand, can arise due to mutations in genes associated with peroxisomal membrane or matrix protein import, resulting in absent or dysfunctional peroxisomes [72]. Zellweger syndrome is a well-known PBD [73], whereas RCDP can result from either single enzyme deficiency or PBDs [74]. Liver abnormality or dysfunction including hepatic steatosis is reported in most of these disorders [75].

### 4.1. Single-Enzyme Deficiencies

#### 4.1.1. Fatty Acid Oxidation-Associated Disorders

Fatty acid oxidation disorders are inherited metabolic conditions occurring due to impaired breakdown of fatty acids [76]. Defective fatty acid oxidation leads to accumulation of toxic lipid intermediates such as pristanic acid, DHCA, THCA, and VLCFAs resulting in abnormal brain development and hepatic dysfunction [77]. Common neurological manifestations include hypotonia, developmental delay, vision and hearing impairments, seizures, peripheral neuropathy, spasticity, and progressive white matter degeneration [78].

X-ALD is the most prevalent peroxisomal fatty acid oxidation disorder (pFAO) caused by mutations in the *ABCD1* gene [79]. This mutation leads to VLCFA accumulation resulting in hepatomegaly and steatosis [80].

D-bifunctional protein (DBP) deficiency is an autosomal recessive disorder caused by mutations in the *HSD17B4* gene [81]. Due to the significant clinical overlap between DBP deficiency and Zellweger syndrome, including liver dysfunction, DBP deficiency is also referred to as pseudo-Zellweger syndrome [82].

ACOX deficiency is mainly caused by mutations in the *ACOX1* gene; however, a few reported cases have also been attributed to mutations in the *ACOX2* gene [83,84]. Clinical manifestations, aside from neurological symptoms include liver fibrosis, and sporadically elevated transaminases [84].

Alpha-methylacyl-CoA racemase (AMACR) is involved in the metabolism of branched-chain fatty acids, particularly pristanic acid, and bile acids [85]. There are two distinct clinical manifestations of AMACR deficiency: early-onset of liver abnormalities (hepatomegaly) and dysfunction (cholestasis) eventuating into liver failure in childhood [86], and progressive cognitive and neurological decline in adult-onset cases [87].

Mutations in the phytanoyl-CoA hydroxylase (*PHYH*) gene can lead to Refsum disease, which is characterised by the accumulation of phytanic acid due to its poor metabolism in peroxisomes [88]. In infantile Refsum disease, belonging to the broader ZSDs, hepatomegaly, liver dysfunction and failure has been observed along with developmental delays [89].

#### 4.1.2. Ether Lipid Synthesis Disorder

RCDP is a peroxisomal disorder caused by mutations in ether lipid (plasmalogen) biosynthesis-associated genes such as alkylglycerone-phosphate synthase (*AGPS*), glycerol-3-phosphate acyltransferase (*GNPAT*), and fatty acyl-CoA reductase 1 enzyme (*FAR1*) [90,91]. RCDP is mainly characterised by profound growth impediment, particularly by the shortening of the proximal limbs (rhizomelia), but also contractures, spasticity, significant developmental delay, and cataracts. [92,93]. While liver dysfunction is not a defining characteristic, some patients may exhibit elevated liver enzymes alkaline phosphatase (ALP), alanine aminotransferase (ALT), aspartate aminotransferase (AST), gamma-glutamyl transferase (GGT), and bilirubin, potentially indicating liver dysfunction [94].

#### 4.1.3. Glyoxylate Peroxisomal Disorder

Mutations leading to a deficiency of alanine-glyoxylate aminotransferase (AGT), encoded by the *AGXT* gene, can lead to primary hyperoxaluria type 1 (PH1) [95]. AGT is crucial for detoxifying glyoxylate in the liver [96]. PH1 is a rare autosomal recessive metabolic disorder associated with the overproduction of oxalate in the liver, which, when combined with calcium to form calcium oxalate crystals, leads to renal damage the hallmark of the disease. A patient with PH1 was detected with surface nodularity and splenomegaly on computed tomography, suggesting liver cirrhosis [97].

### 4.2. Peroxisomal Biogenesis Disorders

#### 4.2.1. PBDs with a Defect in Peroxisomal Membrane and/or Matrix Protein Import

The ZSDs and RCDP types 1 and 5 belong to the largest subgroup of PBDs, which are caused by mutations in any of the 13 different *PEX* genes [98]. Both ZSD and RCDP patients typically present with symptoms of hypotonia, neurological deficits, hearing loss, vision problems, and skeletal abnormalities [99]. Additionally, ZSD are frequently associated with liver dysfunctions, ranging from hepatomegaly and cholestasis to fibrosis, cirrhosis and hepatocellular carcinoma [100]. Importantly, liver disease is a major cause of mortality in these patients [7].

#### 4.2.2. PBDs with a Defect in Peroxisome Division

Peroxisomal fission defects represent a small subgroup of PBDs, arising from mutations in genes responsible for peroxisome division and maintenance [47]. Recent studies have linked these disorders to mutations affecting dynamin 1-like (DNM1L) [101], mitochondrial fission factor (MFF) [102], and PEX11β proteins [103]. Patients with PEX11β deficiency can present with congenital bilateral cataracts and ZSD-like features such as mild progressive hearing loss, intellectual disability, skeletal abnormalities, and sensory neuropathy [104,105]. In contrast, mutations affecting DNM1L and MFF proteins are associated with more severe neurological phenotypes [106,107]. Elevated alanine concentrations in plasma and cerebrospinal fluid-hallmarks of hepatic abnormality, classic symptoms of liver dysfunction have been observed in a case report with a mutation affecting the DNM1L protein [64].

## 5. Liver Disease Associated with Peroxisomal Dysfunction

### 5.1. Liver Disease

The liver is a crucial organ responsible for multiple functions, including processing food nutrients, regulating body metabolism, and removing toxins [108]. Since the liver is responsible for maintaining homeostatic balance and metabolic processes, the development of liver disease can lead to significant patient morbidity and mortality [109]. Symptoms of liver disease include hepatomegaly [110], cholestasis [111], bilirubinemia [112], and fibrosis [113].

Liver disease is responsible for over two million deaths yearly and accounts for 4% of all deaths worldwide [114]. The majority of deaths are due to cirrhosis and hepatocellular carcinoma, with acute hepatitis contributing to a smaller proportion of deaths [115,116]. In Europe, liver disease now ranks as the second leading cause of working life cost, following ischemic heart disease [117]. Similarly, in the United States, cirrhosis-related hospitalizations and corresponding healthcare expenditures, have significantly increased over the past two decades, exceeding those for patients with congenital heart failure [118]. Importantly, two-thirds of the costs related to these conditions are attributable to inpatient or emergency department care. Over the past 20 years, healthcare spending on cirrhosis has increased by 4% per year, primarily driven by hospital-based services [119].

The major contributors of cirrhosis worldwide are related to alcohol, viral hepatitis, and MASLD [120]. Alcohol is one of the most significant causes leading to cirrhosis globally and in high-income countries, its occurrence is even higher [121,122]. Alcoholic liver disease encompasses a spectrum of hepatic disorders, starting with steatosis, potentially advancing to alcohol hepatitis, and subsequently leading to alcoholic cirrhosis [123]. Chronic hepatitis C virus (HCV) and hepatitis B virus (HBV) infections account for 57% of cirrhosis cases globally [124]. It typically takes 10–20 years for viral hepatitis to advance to cirrhosis [125]. During this time, persistent hepatic inflammation caused by the infection leads to repeated cycles of inflammation, necrosis, and regeneration which leads to cirrhosis development [125].

MASLD is the second-leading cause of end-stage liver disease and liver transplantation in Europe and America, affecting a quarter of the global adult population [126,127]. Markov modelling predicted that between 2016 and 2030, the burden of advanced disease due to MASLD would more than double in different regions of the world due to an increase in the rising prevalence of metabolic risk factors and the ageing population [128,129].

MASLD is a broad term given to the range of diseases starting from metabolic dysfunction-associated steatotic liver (MASL) to metabolic dysfunction-associated steatohepatitis (MASH) which can further advance to liver fibrosis, cirrhosis, and hepatocellular carcinoma (HCC) [130]. When lipid droplet buildup increases beyond 5% of the total liver weight, an individual can be considered to have MAFL [131]. When inflammation and fat accumulation combine, around 30% of those with MAFL develop MASH [132]. Due to increasing fibrosis, over 20% of MASH patients may advance to irreversible hepatic cirrhosis [133,134], which impairs liver function and enormously increases the risk of developing hepatocellular cancer (HCC) [135].

The primary diagnostic tool for MASLD is ultrasound despite being less reliable when hepatic steatosis is <20% [136]. Magnetic resonance imaging provides sensitivity; nonetheless, its accessibility is limited [137]. Vibration-controlled transient elastography contributes to the assessment of the fibrosis stage by evaluating liver stiffness [138]. However, cannot differentiate simple steatosis from steatohepatitis, thus liver biopsy remains the gold standard technique, despite it being invasive and expensive [137,139]. This leads to a need to develop a non-invasive tool that could accurately identify MASLD/MASH and classify patients as high and low-risk individuals for fibrosis, as high-risk patients necessitate periodic monitoring and treatment.

Several genome-wide association studies (GWAS) have identified potential genetic biomarkers that may help in the advancement of personalised therapy by improving patient stratification and enabling the application of preventive and therapeutic strategies for MASLD. Many genetic variants linked with the MASLD trait have been identified, further investigation is necessary to detect the causal genes and to understand the molecular pathways by which they contribute to disease progress [140]. Nonetheless, the currently identified genetic variations account for approximately 20% of the total heritability, indicating the presence of yet-to-be-discovered additional risk variants [141].

While MASLD is predominantly associated with obesity, it is also progressively more frequently diagnosed in individuals with normal body weight, with an estimated worldwide prevalence of 4.1%, particularly in Asia [142]. Lean MASLD is thereby identified as mechanistically distinct from obese MASLD, marked by specific clinical and metabolic features [143]. The common factors contributing to the development of lean and obese MASLD include insulin resistance, dyslipidemia, and metabolic syndrome [144]. These factors arise from defective metabolic pathways which include bile acid accumulation [145], impaired ether lipid synthesis [146], peroxisomal fatty acid oxidation [5], and glyoxylate detoxification [147]. The involvement of peroxisomes in these metabolic pathways indicates their potential role in steatosis.

### 5.2. Peroxisomes and MASLD

Peroxisomes play a critical role in managing hepatic lipid metabolism and maintaining ROS homeostasis, processes that are commonly disrupted in MASLD, which is explained in Figure 2. Peroxisomal β-oxidation acts as a sensor for intracellular fatty acids and regulates lipolysis. ACOX1 is the first and rate-limiting enzyme in peroxisomal β-oxidation pathway [148]. Any disturbance in the peroxisomal β-oxidation might lead to hepatic steatosis by toxic lipid accumulation [20]. A supporting study has demonstrated that mutations affecting the ACOX1 protein resulted in MASLD progression which led to significant acceleration of hepatocellular damage by inducing histological changes in hepatocytes, including hepatic inflammation, and upregulation of gene associated with HCC development. This finding suggests the potential importance of peroxisomal β-oxidation in MASLD progression, though further studies are required to understand underlying the mechanism [149].

In a separate investigation, ACOX1 knockout mice demonstrated hepatic metabolic disturbances that precipitated the onset of steatohepatitis, hepatocellular regeneration, spontaneous peroxisome proliferation, and hepatocellular carcinomas [150]. This indicates a possible link between ACOX1 deficiency and liver disease progression, though exact causal mechanisms remain unclear. A study conducted in 2020 showed that liver-specific ACOX1 knockout protected mice against hepatic steatosis caused by starvation or HFD due to induction of autophagic degradation of lipid droplets [151]. However, further studies are required to establish causative mechanisms and assess the significance of human liver disease. A recent study has shown that ACOX2 knockout mice rapidly developed liver cancer with notable lymphocytic infiltrate [152]. While this finding points to the potential regulatory role of ACOX2 in hepatic metabolic homeostasis, further studies are needed to establish mechanistic links and their implication in human liver disease. When exome sequencing was performed on an 8-year-old male presenting liver fibrosis, elevated transaminase levels, mild ataxia, and cognitive impairment, an absence of ACOX2 was observed with an elevation of bile acid intermediate [153]. This finding suggests that ACOX2 loss may be linked with liver dysfunction through disrupted bile acid metabolism; however, further evidence from functional studies is necessary to confirm this relationship. Alterations in ACOX1 and ACOX2 have been implicated in hepatic steatosis and its progression to hepatocellular carcinoma, suggesting a potential association between impaired peroxisomal β-oxidation deficiency and liver pathology. However, the extent to which these enzymes directly contribute to disease progression remains to be fully established. Targeting ACOX1 and ACOX2 could be considered a viable strategy for managing both liver disease and peroxisomal disorders.

Impairment in plasmalogen synthesis leads to disrupted lipid metabolism, elevating oxidative stress and mitochondrial dysfunction, resulting in hepatic lipid accumulation and steatosis [146]. Emerging evidence indicates a depletion of total ether phospholipids in patients with MASLD, suggesting a potential association with peroxisomal deficiency, although additional research is required to determine whether this reflects a causal relationship or a secondary effect of metabolic dysfunction [154]. Dysregulation of peroxisomal enzymes such as AGPS [155] and GNPAT [156] has been shown to lead to the development of hepatocellular carcinoma. These findings suggest that AGPS and GNPAT may modulate hepatic carcinoma cells through lipid-mediated signalling and gene regulation. However, further studies are needed to delineate the direct mechanisms involved and to evaluate these findings in vivo. Administration of the plasmalogen precursor, alkyl glycerol (AG), successfully prevented hepatic steatosis and MASH in a mouse model by increasing fatty acid oxidation [146]. While these findings support a protective role for endogenous in liver lipid metabolism and MASH, further studies are necessary to establish direct causal mechanisms and explore therapeutic potential in vivo. Emerging evidence indicates that ether-phospholipids (ether-PLs) may be linked not only to neurodegenerative disorders but also to steatohepatitis; however, the molecular mechanisms by which ether lipids contribute to these diseases are largely unknown, suggesting a need for future research [157]. Additionally, hepatoprotective effects have been demonstrated by consumption of sea cucumber, a widely consumed dietary item, and traditional remedy in Asia, which is rich in ether-phospholipids [158]. Further studies are required to evaluate translational relevance. Overall, these studies suggest that ether phospholipid synthesising enzymes could be potential targets for treating liver disease.

Glyoxylate accumulation causes mitochondrial dysfunction, oxidative damage, and disrupted lipid metabolism, resulting in accumulation of triglyceride within hepatocytes, leading to steatosis [159,160]. Epigenome and transcriptome analyses of mouse MASLD hepatocytes identified altered glyoxylate metabolism [147]. Hepatocytes isolated from human steatotic livers exhibit hypermethylation and downregulation of AGXT; additionally, severity of steatosis in MASLD adolescents was linked with urinary oxalate excretion [147]. Therefore, defective peroxisomes might reduce the capacity of the fatty liver to detoxify glyoxylate derived from hydroxyproline breakdown, causing overproduction of the harmful waste product oxalate [147]. These findings highlight that disrupted glyoxylate detoxification in the steatotic liver may contribute to increased systemic oxalate levels. However, further validation is needed to confirm causality and translational relevance. In addition, glyoxylate reductase/hydroxypyruvate reductase (GRHPR) was reported as a novel prognostic marker for HCC patients after curative resection [161]. While these findings suggest a potential relation between GRHPR loss and poor prognosis in HCC, further research is needed to elucidate whether GRHPR plays a functional role in tumour progression. Together, these findings point to the emerging significance of glyoxylate metabolism in both MASLD and hepatocellular carcinoma. However, as these associations are observational, further mechanistic studies and clinical studies are necessary to establish causality and evaluate their relevance as prognostic markers or therapeutic targets.

Peroxisomes also play a major role in maintaining this balance between ROS generation and antioxidant defence mechanisms. Peroxisomes produce ROS by taking part in β-oxidation [162]. Peroxisomes also comprise various antioxidant molecules and enzymes to breakdown ROS produced inside and outside the organelle [162]. This implies that peroxisomal gene defects would alter the peroxisomal function of maintaining this balance thereby aggravating the oxidative stress in MASLD/MASH. For instance, a progression of MASLD was observed in a catalase knockout mouse via ER stress [163]. Furthermore, a decreased level of CAT and -262 C/T polymorphism in the promoter of the *CAT* (rs1001179) gene are linked with MAFLD [164], MASH [165], and HCC [166]. Overall, these findings indicate that disruption of peroxisomal redox balance, specifically through catalase loss, may contribute to MASLD and HCC progression via ER stress and oxidative stress. Nonetheless, additional mechanistic and clinical investigations are necessary to confirm causality and to explore catalase or other peroxisomal antioxidant pathways as potential therapeutic targets.

Peroxisomal depletion can occur when peroxisomal membrane proteins or transport proteins are dysfunctional. Steatosis, steatorrhea, and cholestasis were observed in 2-week-old PEX1-G843D knock-in mice [167]; however, the progression of phenotype in the later stage was not available. While this study highlights that PEX1 knock-in may contribute to hepatic dysfunction, additional research is necessary to elucidate the underlying causative mechanisms and to establish precisely the murine phenotype represents human ZSD pathology. In addition, the livers from PEX2 knockout mice showed aggravated steatosis and cholestasis during the first weeks of life [6]. This finding suggests that PEX2 knockout may contribute to steatohepatitis via bile acid–induced cholestasis, necessitating further investigation into its therapeutic importance and underlying mechanisms. Furthermore, a study using mice with liver-specific PEX5 knockouts revealed hepatic steatosis due to peroxisomal deficiency [168]. Although hepatic steatosis is evident in liver-specific PEX5 knockout mice, the translational relevance to human peroxisomal disorders requires further establishment in human-based models. siRNA-mediated knockdown of PEX14 in BRL cells (a rat hepatocyte line), triggered ROS accumulation, lipid buildup, and peroxidation, ultimately resulting in cell death such as apoptosis, ferroptosis, and autophagy [169]. These findings highlight that disrupted PEX14-dependent peroxisome function plays a contributory role in impaired lipid metabolism and cell survival, although confirmation in human models is required to validate translational significance. Additionally, a mouse study has demonstrated that PEX11 loss affects peroxisome elongation and division, increased lipid accumulation, contributing to the development of MASLD [170]. The study highlights that PEX11α deficiency may contribute to hepatic steatosis by disrupting peroxisomal fatty acid β-oxidation and biogenesis, further functional and translational studies are required to validate causality and significance to human disease. These results imply that the peroxisomal depletion caused by peroxisomal gene defects leads to hepatic steatosis and MASH [171].

Peroxisomes also share extensive metabolic connections with other cell organelles. A 2022 study found that in diabetic mice, excessive oxidation of fatty acids by peroxisomes due to impaired mitochondrial FAO in the liver contributes to hepatic lipid accumulation [5]. Peroxisomal β-oxidation may influence mitochondrial fatty acid metabolism in obesity and diabetes mouse model; but further research is necessary to confirm its clinical relevance in human disease. Another study showed that peroxisomal depletion leads to cholesterol accumulation in lysosomes playing a significant role in development of inflammation responsible for MASH [172]. Accumulation of cholesterol within lysosomes appears to play a contributory role in driving inflammation in metabolic diseases such as MASH; however, additional research is needed to fully understand the mechanistic pathways and to validate clinical relevance of targeting lysosomal cholesterol in human models. In summary, these discoveries highlight an unforeseen function of peroxisomes in MASLD/MASH.

Hepatic dysfunction was reported in the PBDs mentioned above. Histological analysis of biopsies from Zellweger syndrome patients revealed advanced intralobular and periportal fibrosis, along with micronodular cirrhosis due to accumulation of bile acid intermediates like DHCA and THCA [173]. However, additional research is needed to confirm their causative role and potential therapeutic strategies. Similarly, patients with ZSDs such as neonatal adrenoleukodystrophy and infantile Refsum disease frequently presented with liver damage progressing in cirrhosis; however, the underlying mechanisms remain unclear and require further study. [174,175]. In a mouse model of Refsum disease (*Phyh*^−/−^), phytol supplementation led to formation of large lipid vacuoles in the liver parenchyma, with steatosis progressing from micro to macrovesicular forms in a dose-dependent manner [176]. The *Phyh*^−^/^−^ mouse model shows that phytanic acid accumulation induces hepatic steatosis, although the translational significance to human pathology and potential therapeutic strategies still require additional investigation. Additionally, ZSD patients have higher risk of developing HCC, as evidenced by a case report of a 36-year-old developing HCC and precancerous lesions in an 18-year-old, suggesting the relevance of liver assessment in ZSD management [7]. However, conclusions are restricted by the small cohort size.

A clinical case of defective peroxisomal β-oxidation due to ABCD3 deficiency presented the onset of hepatosplenomegaly in a patient at the age of 1.5 years. At the age of five, their liver enlarged and became cirrhotic [177]. These findings suggest a potential role of mutations in the ABCD3 protein in liver dysfunction, although validation across diverse experimental models and additional clinical cases is necessary to generalise these results. In older patients with alanine/glyoxylate aminotransferase-1deficiency, mild to moderate fibrosis was observed; however, long-term exposure to oxaluria might lead to liver cirrhosis [97]. This case shows the potential link between prolonged oxalate accumulation in PH1 and liver cirrhosis, warranting further research to confirm its clinical significance. ABCD2 knockout mice, a mouse model of X-ALD patients, developed hepatic steatosis when fed an erucic acid (C22:1) enriched diet [178]. This finding suggests that ABCD2 protects against erucic acid–induced hepatic steatosis, although further studies are necessary to establish its role across diverse dietary and physiological models. Collectively, these findings underscore the essential role of peroxisomal integrity in maintaining liver function and preventing the progression of hepatic steatosis.

## 6. Conclusions

Peroxisomes play a central role in many major metabolic pathways in the liver, such as fatty acid oxidation, maintaining ROS balance, glyoxylate detoxification, synthesis of ether lipid, cholesterol, and bile acid. These functions are facilitated through its dynamic interaction with various other organelles such as ER, mitochondria, and lipid bodies. The repercussions of partial or complete peroxisome dysfunction can impact these crucial metabolic pathways and the functioning of other organelles which can disrupt lipid homeostasis and ROS metabolism in the liver. Patients with PBDs exhibit a range of liver dysfunction, including mild to severe steatosis, fibrosis, and cancer.

MASLD affects at least a quarter of the global adult population. This condition reflects fat accumulation in the liver due to metabolic risk drivers and without excessive alcohol consumption. Fat accumulation can occur when there is reduced fatty acid oxidation, increased fatty acid synthesis, or a combination of both. Peroxisomes are crucial in fatty acid oxidation and fatty acid synthesis processes. This suggests a potential association between peroxisomal defects and liver dysfunction like MASLD.

Recent evidence from both clinical observations and experimental models highlights a clear association between peroxisomal dysfunction and liver pathology, although the precise molecular mechanisms involved are not clear. Integrating single-cell and spatial transcriptomics with functional metabolomics could reveal how peroxisome-associated pathways influence liver disease. Future basic research should investigate how interactions between peroxisomes, ER, and mitochondria contribute to liver dysfunction and aim to identify the molecular pathways connecting these organelles to disease progression.

Targeting peroxisomal pathways might present novel therapeutic opportunities for liver diseases. Gene therapy approaches and CRISPR-based correction of pathogenic *PEX* gene mutations offers a promising therapeutic strategy for treating peroxisome-related liver diseases by directly restoring peroxisomal function at the genomic level. Early diagnosis of peroxisome-associated liver disease through biomarkers such as bile acid intermediates or oxalate levels may enhance patient outcomes. Long-term, personalised approaches integrating dietary interventions (e.g., reduced phytanic acid intake), antioxidant treatment, and potential gene replacement therapies could transform the clinical management of both rare (e.g., ZSD, Refsum disease) and common metabolic liver diseases associated with peroxisomal dysfunction.

In this review, we have detailed the consequences of peroxisomal dysfunction on the pathogenesis of MASLD.

## Figures and Tables

**Figure 1 ijms-26-08303-f001:**
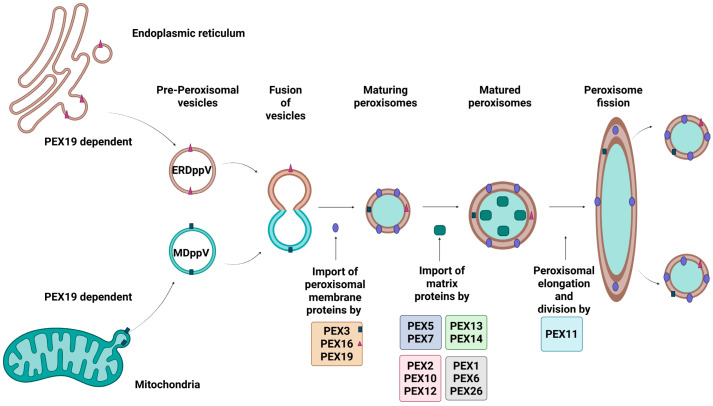
Peroxisomal Biogenesis in Mammalian Cells. The figure illustrates key steps involved in peroxisomal biogenesis. Peroxisomal membrane proteins (PMPs) are synthesised in the cytosol and are recognised by PEX19, which delivers them to the peroxisomal membrane by interacting with PEX3 and PEX16, mediating their insertion. Matrix proteins containing PTS1 or PTS2 signals are identified by cytosolic receptors PEX5 and PEX7, respectively, and targeted to the peroxisome. The receptor-cargo complex dock at the membrane via the PEX13–PEX14 complex, allowing cargo translocation into the matrix. After releasing the cargo, PEX5 is monoubiquitinated by the PEX2–PEX10–PEX12 complex and removed from the membrane by the PEX1–PEX6 ATPase complex, which is anchored by PEX26, allowing recycling of receptor back into the cytosol, ensuring proper peroxisome assembly and maintenance. In addition to import, peroxisome elongation and division is regulated by PEX11. Created in Bio Render. Subramaniam, V.N. (15 June 2025) https://BioRender.com/uyr6yts.

**Figure 2 ijms-26-08303-f002:**
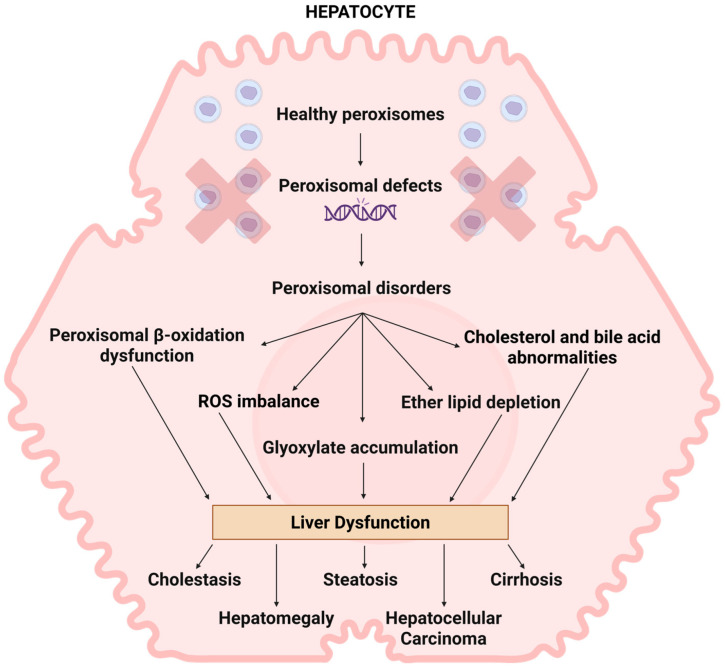
The role of peroxisomes in liver disease. The presence of peroxisome biogenesis dysfunction commonly impacts the liver in patients. The manifestations of liver pathology exhibit significant variability and are closely linked to the disease’s severity. The deficiency of peroxisomes results in impaired peroxisomal beta-oxidation, ROS imbalance, accumulation of glyoxylate, cholesterol and bile abnormalities, and reduction in ether lipids. Studies indicate that these irregularities may contribute to liver dysfunctions like cholestasis, hepatomegaly, steatosis, cirrhosis, and hepatocarcinoma; however, the exact mechanisms behind these occurrences remain unclear. Created in Bio Render. Subramaniam, V.N. (15 June 2025) https://BioRender.com/uyr6yts.

## Data Availability

No new data was created.

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
