# Peer review of "Peroxisome Dysfunction and Steatotic Liver Disease"

_ijms, 2025, doi:10.3390/ijms26178303_

Round 1

Reviewer 1 Report

Comments and Suggestions for Authors

 This review aims to summarize the roles of peroxisomes in hepatic metabolism and their potential involvement in the pathogenesis of metabolic dysfunction-associated steatotic liver disease (MASLD). Overall, the manuscript is comprehensive and well-structure. However, it still needs revision before publication.

1.The topic involves two parts:“peroxisome” and “liver disease”,so I think in the introduction part, authors should also introduce the relationship between clinically diagnosed peroxisome abnormalities and fatty liver disease.This will better stimulate readers' interest and make the significance of basic research greater.

2.I don't understand why the author spent so much space on the section on peroxisome abnormalities in the fourth part, and this part has no corresponding relationship with the content in section 5.2. According to the title, the author mainly introduces the abnormality of peroxisomes in fatty liver disease. If arranged according to the current structure, should the author adjust the title? Or compress the content of the fourth part.

3.The manuscript often presents correlations (e.g., between gene mutations and liver pathology) without sufficiently elaborating on whether these represent causative relationships or merely associations. This is especially relevant when extrapolating findings from rare genetic diseases or animal models to common MASLD cases.

4.The manuscript does not address the heterogeneity within MASLD populations (e.g., lean vs obese MASLD), nor does it explore how peroxisomal dysfunction may differentially contribute to these subtypes.

5.Please added the future perspective, including both basic research, drug development and possible clinical treatment.

Reviewer 2 Report

Comments and Suggestions for Authors

The manuscript by Vinoy et al. addresses a topic of potential clinical relevance. It aims to highlight the involvement of peroxisomal defects in the pathogenesis of MASLD by conducting a comprehensive review of the subject.

Manuscript could be improved using more figures to illustrate the summarized reviewed information throughout the whole manuscript, starting from the introduction section, for example, adding a figure summarizing peroxisome biogenesis. Also, a graphical abstract could be added.

Regarding peroxisomal disorders epidemiology, could authors provide incidence and estimated prevalence data of peroxisomal genetic diseases mentioned at the manuscript, if available?

Regarding references, of the 201 references cited, approximately 25% of them are publications within the last 5 years. How authors address the fact of not having included many recent publications in this review?

Regarding reference 2, it seems incomplete and mainly related to kidney

J.1954 R. Correlation of ultrastructural organization and function in normal and experimentally changed proximal tubule cells 437 of the mouse kidney 1954. 

Regarding reference 97

Wang N, Kong R, Luo H, Xu X, Lu J. Peroxisome Proliferator-Activated Receptors Associated with Nonalcoholic Fatty Liver613Disease. PPAR Res.2017;2017:6561701,

PPAR have not been cited throughout the manuscript

Conclusions statements are coherent but as authors reported, molecular mechanisms are not clear and further research is needed.

Have the authors ever considered the possibility of conducting a systematic review on this topic? Undertaking a standardized methodology, following a predefined protocol and search strategy would strengthen the reliability of the findings and facilitate reproducibility. This might provide a more comprehensive overview of the existing evidence.

Line 13 and bony stippling) in the patellae

Line 391 neonatal adrenoleukodystrphy

Use of both terms glyoxylate and glyoxalate (text and figure 1)

Round 2

Reviewer 1 Report

Comments and Suggestions for Authors

no more suggestion.